# Transforming medical device biofilm control with surface treatment using microfabrication techniques

**Mohammad T. Al Bataineh**[ID][1]*, **Anas Alazzam**[ID][2]*

**1** Center for Biotechnology, Department of Molecular Biology and Genetics, College of Medicine and Health Sciences, Khalifa University, Abu Dhabi, United Arab Emirates, **2** System on Chip Lab, Department of Mechanical Engineering, Khalifa University, Abu Dhabi, United Arab Emirates

* mohammad.bataineh@ku.ac.ae (MTA); anas.alazzam@ku.ac.ae (AA)

**Data Availability Statement:** All relevant data are within the paper.

**Funding:** The author(s) received no specific funding for this work.

## Abstract

Biofilm deposition on indwelling medical devices and implanted biomaterials is frequently attributed to the prevalence of resistant infections in humans. Further, the nature of persistent infections is widely believed to have a biofilm etiology. In this study, the wettability of commercially available indwelling medical devices was explored for the first time, and its effect on the formation of biofilm was determined *in vitro*. Surprisingly, all tested indwelling devices were found to be hydrophilic, with surface water contact angles ranging from 60˚ to 75˚. First, we established a thriving *Candida albicans* biofilm growth at 24 hours. in YEPD at 30˚C and 37˚C plus serum *in vitro at Cyclic olefin copolymer (COC) modified surface*, which was subsequently confirmed via scanning electron microscopy, while their cellular metabolic function was assessed using the XTT cell viability assay. Surfaces with patterned wettability show that a contact angle of 110˚ (hydrophobic) inhibits *C. albicans* planktonic and biofilm formation completely compared to robust growth at a contact angle of 40˚ (hydrophilic). This finding may provide a novel antimicrobial strategy to prevent biofilm growth and antimicrobial resistance on indwelling devices and prosthetic implants. Overall, this study provides valuable insights into the surface characteristics of medical devices and their potential impact on biofilm formation, leading to the development of improved approaches to control and prevent microbial biofilms and re-infections.

## Introduction

Indwelling medical device-associated infections caused by biofilms represent a serious threat to public health [1]. Biofilms that form on medical devices are difficult to control with antimicrobials due to innate and developed mechanisms of resistance, physiological gradients, and matrix diffusion limits, all of which promote antimicrobial resistance [1, 2]. Multiple multi-drug-resistant microorganisms were isolated from the most prevalent implantable devices, such as *Staphylococcus aureus*, *Pseudomonas aeruginosa*, *Enterococcus*, and *Acinetobacter baumannii* [3]. In lieu of treating the developed biofilm, coating or surface modification to prevent the attachment of microorganisms is a new method that has been intensively studied to

**Competing interests:** The authors have declared that no competing interests exist.

**Abbreviations:** COC, Cyclic olefin copolymer; ICU, Intensive Care Unit; OD, Optic Density; XTT, 2,3-Bis (2-Methoxy-4-Nitro-5-Sulfophenyl)-2H-Tetrazolium-5-Carboxanilide; YEPD, Yeast Extract Peptone Dextrose.

combat the problem of biofilm development in medical devices [4]. According to the National Institute of Health, biofilms are responsible for a significant proportion of microbial infections in humans, including persistent infections related to implantable and indwelling devices. This can account for up to 80% of the total number of such infections [34]. Biofilms on indwelling devices serve as infection and re-infection niches. Biofilms are intricate assemblages of micro-organisms that possess significant biological capabilities, such as heightened resilience to external stressors such as antimicrobial agents. The process of biofilm formation is widely recognized to be heavily influenced by the characteristics of the substrate, which encompass a range of properties. The phenomenon of (macro) molecules being adsorbed onto a substrate, commonly referred to as a conditioning film, has been observed to alter the physicochemical characteristics of the surface, ultimately influencing the adhesion of bacteria [4].

The utilization of implanted medical devices in contemporary therapeutic interventions is linked to a significant proportion of nosocomial and systemic infections, with *Candida albicans* infections being the fourth most prevalent cause of nosocomial bloodstream infections [5, 6]. Fungal infections are frequently linked with Candida spp., with *C. albicans* being the predominant species responsible for both superficial and systemic ailments [7, 8].

*C. albicans* is a significant fungal pathogen that affects humans and causes localized and systemic candidiasis, including bloodstream infections that are acquired in hospital settings, especially among immunocompromised individuals [9]. The treatment of *C. albicans* infections has become increasingly challenging owing to a restricted repertoire of antifungal medications and the heightened prevalence of drug-resistant isolates, which has led to an increased mortality rate associated with candidiasis [10, 11]. The virulence of *C. albicans* is contingent upon its capacity to undergo a reversible transformation from yeast cells, which are single ovoid cells that bud, to elongated cells that are attached end-to-end and form pseudohyphal and hyphal filaments [12]. This transformation is triggered by host environmental cues such as serum, pH, and body temperature (37˚C). The aforementioned transition facilitates the effective infiltration of *C. albicans* into tissues, including the oral mucosa, while also enabling immune system avoidance and subsequent spread. The formation of *C. albicans* biofilm is a crucial virulence characteristic that involves a shift from a solitary state to a multi-layered configuration that is linked to a surface. This configuration is enclosed within an extracellular matrix (ECM) that is rich in polysaccharides and is known to exhibit increased resistance to antimycotic drugs [13, 14]. Several *Candida* species have been investigated in relation to biofilms; however, *C. albicans* has been the subject of the most comprehensive research.

Biofilm formation, among other biological processes, depends on the contact of cells with surfaces, and cell adhesion is a crucial component of this interaction [15]. The wettability of the surface is one of the crucial elements that govern cell attachment [16]. Surface functional groups and texture have an impact on the surface's ability to cling to liquids [17]. This characteristic, in turn, influences how cellular proteins adhere, which significantly impacts how cells behave. The prevalent method employed for evaluating the wettability of a polymer is to ascertain the water contact angle on its surface. According to research, cells typically stick to surfaces with touch angles between 40 and 70 degrees [18]. It's crucial to remember that the effect of wettability on cell activity can change based on particular surface characteristics. Superhydrophobic surfaces have been observed to induce a conformational change in fibronectin, resulting in a decrease in cell adhesion. Fibronectin is a major mechanism for adhesion to the fungal cell wall. Conversely, hydrophilic surfaces are shown to stimulate cell division [19]. Therefore, when designing biomaterials for indwelling medical devices and other uses that involve cell interactions with surfaces, surface wettability is a crucial element to consider. Modifying the surface chemistries of biomaterials, such as by introducing surface-modifying end groups (SMEs) or changing the chemical makeup of substrates, is another strategy for

preventing or reducing the formation of biofilms on them. One study looked at how the presence of SMEs influenced *C. albicans*' capacity to build biofilm [20].

The choice of substrates employed in biological assays holds a paramount influence on the overall performance and results of the tests. These substrates have the potential to impact critical factors such as cell viability, growth, and the adsorption of proteins and drugs [21]. Cyclic olefin copolymer (COC) has emerged as a notable material in the field of biotechnology due to its outstanding characteristics. Notably, has emerged as a highly promising material due to its typically low or negligible extractable properties, rendering it exceptionally biocompatible [22].

In this study, the main objective is to investigate the formation of *C. albicans* biofilm on surfaces with varying levels of wettability. Furthermore, we measured the wetting properties of commercially accessible indwelling medical apparatus. The rationale for incorporating these measurements into the current study is to gain insights into the wettability of their internal surfaces. The findings of the research can offer significant perspectives on the variables that contribute to the development of biofilm on medical apparatus. The study may also aid in the development of infection control strategies to inhibit biofilm growth, which is a main challenge in healthcare settings.

## Methods

### Strains and culture conditions

A wild-type *C. albicans* (DK318) strain was used throughout this study. Standard growing conditions that do not induce filament formation were employed in this study: the YEPD medium, which consisted of 2% yeast extract, 2% peptone, and 1% glucose and was maintained at a temperature of 30˚C. The procedure for the biofilm formation assay was conducted following established protocols, as previously outlined [23]. In short, the wild-type variant was cultured in YEPD medium at 30˚C with an OD600 of approximately 4.0. The culture was then diluted at a ratio of 1:10 into 50 ml of pre-warmed YEPD medium supplemented with 10% fetal bovine serum (FBS) at 37˚C. The resulting cultures were agitated at 200 rpm for a duration of 24 hours, following previously established protocols [24]. For imaging purposes, the cultures were collected at a specific time point of 24 hours, followed by fixation using 4.5% formaldehyde. Subsequently, they were washed twice with 1× phosphate-buffered saline (PBS) to eliminate any remaining YEPD for imaging purposes [25].

### Biofilm formation

The study quantified the biofilm growth levels through a standard, semi-quantitative colorimetric XTT reduction assay. Briefly, a wild-type *C. albicans* strain cell suspension ($1 \times 10^7$ cells/ml) was applied on a COC surface featuring patterned wettability (contact angle of 40˚ and 110˚). The strain was cultivated for one night in YEPD medium under control conditions (non-filamentous) at 30˚C and in YEPD medium supplemented with 10% serum under strong filament-inducing conditions at 37˚C and a standardized *C. albicans* to the surfaces placed in a tissue culture plate. The XTT reduction assay, also known as -2H-tetrazolium-5-carboxanilide assay, is a commonly used method in research to measure cell viability and proliferation [26]. For the XTT assay, we used two separate (hydrophobic and hydrophilic) treated after 24 hours. of growth, accordingly.

## Microfabriation for modifying surface wettability

Cyclic olefin copolymer (COC) is a highly advantageous polymer extensively employed in various biological applications [21]. Its exceptional properties, such as transparency, stiffness, surface treatment stability over time, and resistance to several solvents and acids, make it an ideal choice for a variety of biological and biotechnological applications. The wettability of the COC substrate surface was altered through a standard microfabrication process, which is illustrated in Fig 1 [27]. Initially, the substrate was cleansed by immersing it in an isopropanol bath for five minutes, followed by DI water for an additional five minutes. Subsequently, compressed nitrogen was used to dry the substrate, which was then baked at 70 degrees Celsius for ten minutes. Subsequently, the substrate was coated with a layer of positive photoresist (PR 1813) through the utilization of a spin coater (WS650Hzb-23NPP UD-3 from Laurell Technologies Corporation, North Wales, PA, USA). Afterward, the wafer was soft baked and patterned using a photolithography system with a 10 μm laser beam (Dilase 650 from KLOE, France). The COC substrate was exposed to an appropriate developer and sonicated in DI water, followed by drying with compressed nitrogen.

The patterned portion of the substrate was then treated with oxygen plasma (PDC-002 from Harrick Plasma, USA) for 30 seconds at 700 mTorr pressure and 7.2 W power to modify its wettability. The photoresist layer was removed by sonicating the substrate in an acetone bath. As a result, the exposed surface of the substrate became hydrophilic, while the surface covered by the photoresist layer remained hydrophobic. The COC substrate was entirely cleaned in a bath of DI, dried using compressed nitrogen, and sanitized with isopropanol before attaching it to a 10 ml petri dish using PDMS.

To analyze the surface properties of the substrate, contact angle measurements were conducted using a contact angle goniometer (L2004A1 from Ossila). Prior to the measurement, droplets of 5 μl volume were meticulously deposited on the surface using a pipette. The sessile droplet was then photographed and analyzed using the Ossila software. Multiple measurements were conducted at various locations on the substrate to obtain an accurate representation of the surface properties. The mean value and standard deviation values were subsequently calculated and reported. Fig 2 depicts the contact angles of two droplets placed on COC and plasma-treated COC, respectively. The surface of the bare COC exhibited a

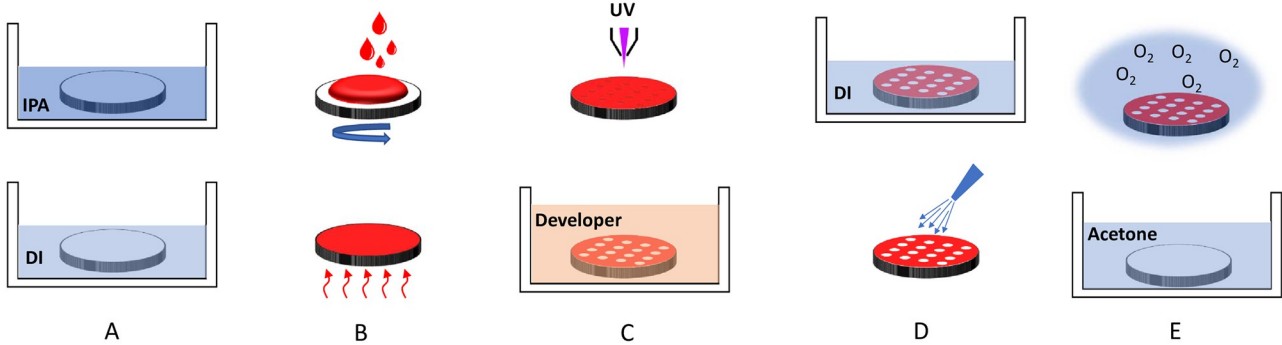

**Fig 1. Schematic diagram for the process used to fabricate a wafer with patterned wettability.** The process involved five key steps: (A) Cleaning the substrate using Isopropanol and DI baths; (B) Depositing photoresist using a spin coater and baking; (C) Patterning the photoresist layer using a photolithography system and developing the photoresist; (D) Cleaning the wafer in DI baths followed by drying using compressed air; (E) Treating the wafer with $O_2$ plasma and performing liftoff using acetone. The COC surface exhibits hydrophobic properties overall, except for hydrophilic regions confined within circular patterns on its surface.

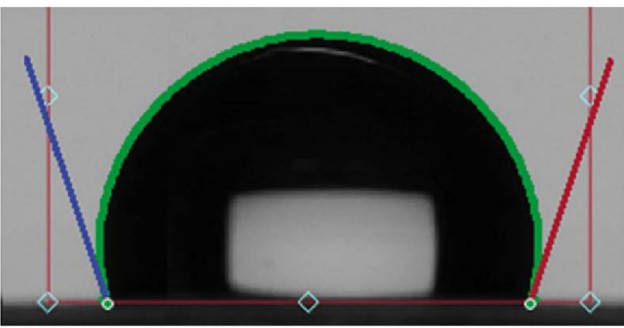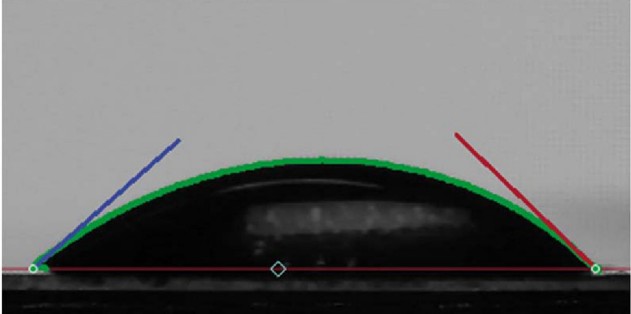

**Fig 2. The contact angle of water sessile droplet on (A) COC and (B) plasma-treated COC.**

contact angle of 110˚±3, whereas the surface of the plasma-treated COC displayed a significantly lower contact angle of 40˚±3, as shown in Fig 2.

### Statistical analyses

Each experiment was run three times with triplicates each time. A *P* value of less than 0.05 was regarded as statistically significant for all statistical tests (unpaired *t* test and correlation coefficient).

## Results

### Water contact angles of commonly used indwelling medical devices revealed hydrophilic surface energy

To investigate the influence of the surface energy, wettability, of indwelling medical devices on the formation of *C. albicans* filament, we conducted a series of contact angle measurements on several commercially available devices. Four indwelling devices were selected for this study: The Butterfly Cannula, Thoracic Catheter, Ryles Tube, and Foley Catheter. Contact angle measurements were performed on the interior part of each device to determine their wettability properties. In order to measure the contact angle, each device was carefully cut and attached to a glass substrate using double-sided tape to ensure a flat and stable surface. Droplets of 5 μL were then placed on the surface of each device, and utilizing a contact angle goniometer, the contact angle was measured. The resulting data for each device was analyzed and presented in Fig 3. The results of our study revealed that all four tested devices exhibited hydrophilic surfaces with contact angles within the range of 60 to 75 degrees, irrespective of their manufacturing material.

### *In vitro C. albicans* biofilms formation

We tested *C. albicans* biofilm formation *in vitro* on 96-well polystyrene plates as shown in Fig 4. We observed increased filamentation encased within the ECM when cells were treated with serum at 37˚C (Fig 4B), compared to ECM-coated yeast cells at 30˚C (Fig 4A).

### *C. albicans* biofilms distinctly flourished on surface patterned wettability *In vitro*

In order to investigate the impact of wettability on the biofilm formation of *C. albicans*, we designed a study where we patterned the surface of COC wafers to feature two distinct

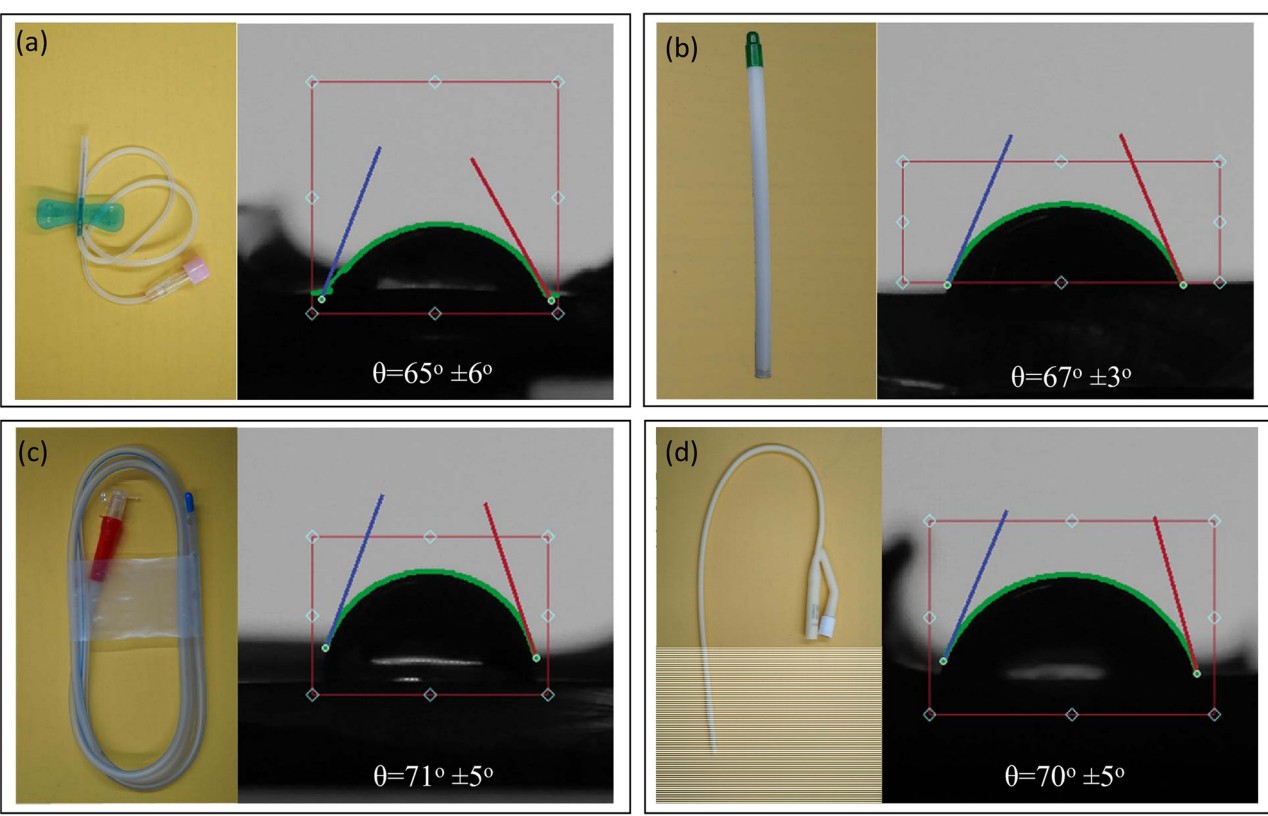

**Fig 3. Four commercially available indwelling devices and the contact angle on the interior surface of the device.** Butterfly Cannula made from PVC and sterilized using irradiation (A), Thoracic Catheter made from clear thermosensitive PVC and sterilized using Ethylene oxide gas. (B), Ryles tube made from medical grade PVC and sterilized by Ethylene oxide gas (C), and Foley Catheter made from latex and sterilized by Ethylene oxide gas (D).

wettabilities using microfabrication and plasma treatment to selected surface areas. Specifically, we created hydrophilic circles on top of the hydrophobic bare COC surface. This experimental design will allow us to observe and analyze the relationship between wettability and biofilm formation in *C. albicans*. Fig 5A shows a photograph of the COC wafer with patterned wettability after being washed in deionized (DI) water, revealing a distinctly patterned wettability of the surface. The hydrophilic regions, which were plasma-treated, exhibit water droplets that stick to the surface, while the untreated hydrophobic areas show no droplets. This clear visual evidence confirms the wettability patterning of the surface, which was later utilized to culture *C. albicans*. Fig 5B presents a photograph of the COC wafer with *C. albicans* after being cultured for 24 hours in a YEPD medium. The results show that *C. albicans* growth occurred mainly on the plasma-treated surface of the COC. The growth is visible to the naked eye and follows the wettability pattern of the surface. A 100X zoomed image of the *C. albicans* growth on the patterned surface of the COC is presented in part 5C of the figure. The results in Fig 5 demonstrate the effect of surface wettability on promoting *C. albicans* growth.

### Surface wettability inhibits *C. albicans* biofilms attachment *In vitro*

We noticed a significant inhibition in the cells' ability to attach to the hydrophobic surface, as shown in Fig 6.

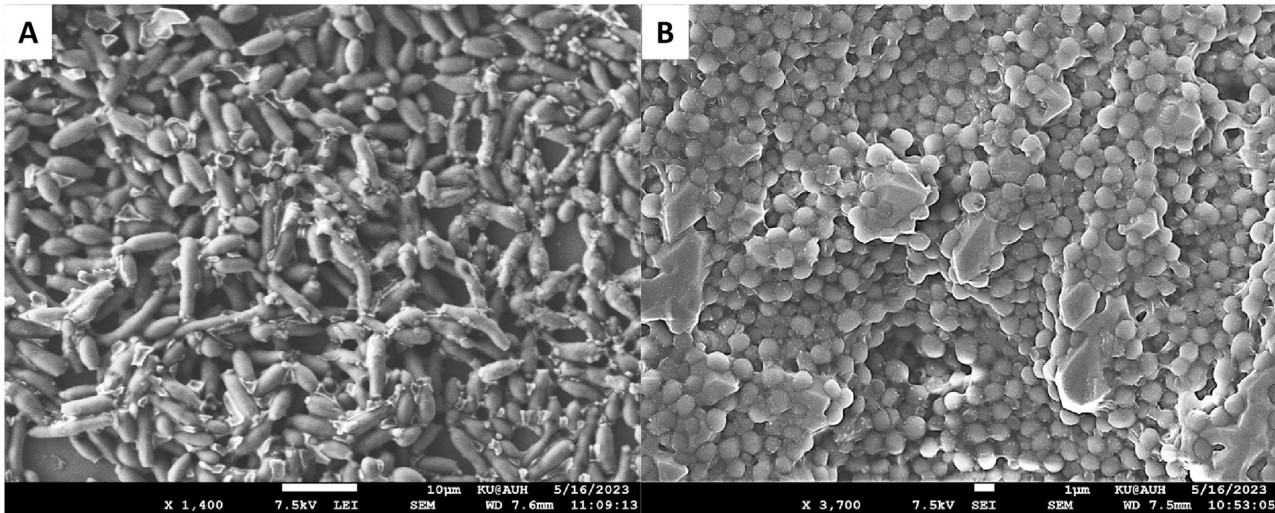

**Fig 4.** *In vitro C. albicans* **biofilm formation for 24 hours. at 30°C (A) and 37°C plus 10% FBS (B) on 96-well polystyrene plates in YEPD media.**
The *C. albicans* strain of wild type was cultured in YEPD medium at 30°C, under non-filament-inducing conditions, for an overnight period. Subsequently, it was diluted at a ratio of 1:10 into pre-warmed YEPD medium that contained 10% serum, and cultured at 37°C, under strong filament-inducing conditions. At a 24 hours time point, the cells were collected, treated with 4.5% formaldehyde for fixation, and subsequently rinsed twice with 1× phosphate-buffered saline (PBS). The imaging process involved the utilization of a scanning electron microscope, with a magnification of 3.00 kx and an acceleration voltage of 15.0 kV.

Attachment is a critical step in the pathogenesis of *C. albicans*. Biofilms were grown at both non-filament-inducing conditions (30°C) and strong filament-inducing conditions (37°C plus serum). *C. albicans* growth was exclusively visible within the hydrophilic region after washing, as shown in Fig 6A. Next, we assessed biofilm growth levels using a standard colorimetric XTT reduction assay as shown in Fig 6B. While we observed an expected growth difference between

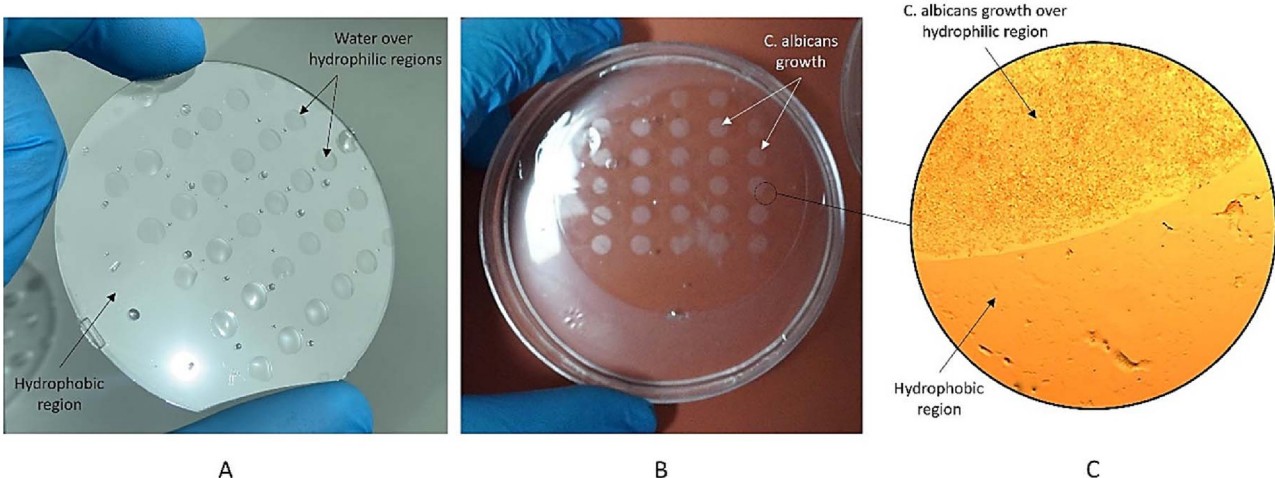

**Fig 5.** (A) the COC wafer with patterned wettability after being washed in DI. Water droplets appear on the hydrophilic regions (B) the COC wafer with *C. albicans* after being cultured for 24 hours. C.albicans growth is observed in the hyrpophilic regions And (C) a 100X zoomed image of the growth of *C. albicans* on top of the patterned surface. The COC surface exhibits hydrophobic properties overall, with the exception of hydrophilic regions confined within circular patterns on its surface.

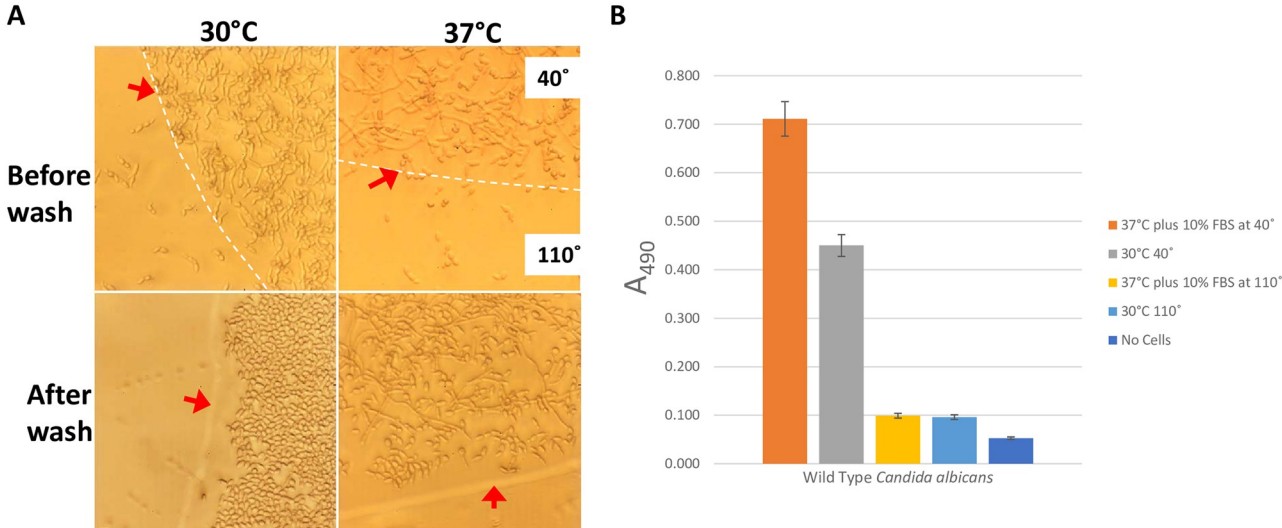

**Fig 6. *In vitro* establishment of *C. albicans* biofilm for a duration of 24 hours.** The experiment involved the cultivation of a wild-type *C. albicans* strain on a COC surface featuring patterned wettability (contact angle of 40˚ and 110˚). The strain was cultivated for one night in YEPD medium under control conditions (non-filamentous) at 30˚C and in YEPD medium supplemented with 10% serum under strong filament-inducing conditions at 37˚C (A). Cells were washed twice with 1× phosphate-buffered saline (PBS), as shown in 'After wash'. The boundary between the hydrophilic and hydrophobic regions is indicated by a red arrow. Notably, the red arrow is located within the hydrophobic region. Microscopy images show that *C. albicans* growth is exclusively visible within the hydrophilic region. Scalebar is 10 μm in length. The evaluation of biofilm formation was conducted through the utilization of a conventional colorimetric XTT reduction assay (B). The error bars in the graph are indicative of the standard errors.

cells growing at 30˚C and 37˚C plus serum, both conditions displayed minimal XTT assay reads on the hydrophobic surfaces and were comparable to the control with no cells.

## Discussion

Attachment and adhesion are critical steps in the pathogenesis of almost all microorganisms, allowing the microbes to attach to host surfaces and adhere to each other, forming biofilms that can persist in host defenses and resist antimicrobial therapies, especially on indwelling devices and implants [28]. Understanding the mechanisms of surface attachment and related biophysical characteristics is crucial for the development of new antimicrobial strategies to prevent resistance and biofilm formation.

In this study and for the first time, we explored the wettability of commercially available indwelling medical devices and determined their effect on biofilm formation *in vitro*. First, we determined the water contact angle for the interior lumen of the Butterfly Cannula, Thoracic Catheter, Ryles Tube, and Foley Catheter. Surprisingly, all four tested devices revealed hydrophilic surfaces with contact angles in the range of 60˚–75˚. While we explored surface energy for the first time, other studies have reported limited success with biochemical coating techniques to mitigate biofilm formation. For example, commercial catheters were coated with antimicrobial agents to interfere with the attachment and expansion of biofilms, and one study coated the ureter polyurethane stents with heparin to prevent biofilm development compared with uncoated stents *in vivo* for 6–8 weeks [29–31]. One limitation of this technique is the naturally increased resistance of various microbes and the dynamic change in the microbial spectrum to the coated antimicrobial agents. Other mechanical techniques employ various types of acoustic energy to eliminate biofilms, such as high-energy ultrasound, which was found to be

limited to certain types of indwelling devices depending on the power intensity doses for each device [32].

We further examined the manufacturer instructions for the tested devices and discovered that sterilization using irradiation is the standard method used. From the literature, it has been noticed that gamma-irradiated surfaces exhibit an increase in wettability, which is consistent with our results in Fig 3 [33, 34]. While this electrochemical phenomenon is still not well understood, one study observed a radiolysis-induced increase in local oxidation and porosity that is directly related to contact angle variation [35]. Next, to investigate the growth of *C. albicans* on COC surfaces with patterned wettability, we conducted *in vitro* XTT reduction experiments. The surface wettability of the COC wafer used in this study was patterned as outlined in Fig 1.

Briefly, the surface of the COC wafer was carefully patterned to include predefined hydrophilic areas (with a contact angle of 40˚) as well as hydrophobic areas (with a contact angle of 110˚) [22]. Hydrophilic areas of the patterned wettability inhibited *C. albicans* biofilm formation completely, growing in YEPD medium at 30˚C and YEPD medium plus 10% serum at 37˚C as shown in Fig 6. These findings suggest that the wettability properties of indwelling medical devices do impact the formation of *C. albicans* biofilm. Further confirmation of biofilm development was provided by electron microscopy and the XTT reduction assay. Interestingly, hydrophobic areas of patterned wettability did not affect filament formation, but they significantly affected *C. albicans* cells' ability to attach to the treated surfaces. This new finding may provide a new antimicrobial strategy to treat indwelling devices and prosthetic implants to overcome biofilm formation and antimicrobial resistance.

Candidiasis ranks as the fourth most prevalent reason for bloodstream infections acquired during hospitalization, with a particular prevalence among individuals who are immunocompromised or receiving care in intensive care units [36]. In this study, we used *C. albicans* biofilms as a relevant clinical model, but we acknowledge the need to validate our findings on mixed microbial biofilms and also test them *in vivo* on several indwelling devices and prosthetic implants. We also plan to test for XTT biofilm formation on the indwelling medical device coating in future studies.

Overall, the current study aimed to determine the hydrophobicity of commercially available indwelling devices and found them to have hydrophilic surfaces, with surface water contact angles ranging from 60˚ to 75˚. Next, we investigated *C. albicans* biofilm formation on surfaces with varying levels of wettability and provided a new insights into the surface properties of indwelling medical devices and their potential impact on biofilm formation. By better understanding the wettability characteristics of these devices, we are likely to provide novel information leading to the development of improved and more effective surface treatment strategies to potentially control and prevent microbial biofilm formation and re-infections.

## Acknowledgments

The authors would like to thank David Kadosh (UT Health, San Antonio, United States) for providing *C. albicans* strains.

## Author Contributions

**Conceptualization:** Mohammad T. Al Bataineh.

**Data curation:** Mohammad T. Al Bataineh.

**Formal analysis:** Mohammad T. Al Bataineh, Anas Alazzam.

**Investigation:** Mohammad T. Al Bataineh, Anas Alazzam.

**Methodology:** Mohammad T. Al Bataineh, Anas Alazzam.

**Project administration:** Mohammad T. Al Bataineh.

**Resources:** Anas Alazzam.

**Software:** Mohammad T. Al Bataineh, Anas Alazzam.

**Supervision:** Mohammad T. Al Bataineh.

**Validation:** Mohammad T. Al Bataineh, Anas Alazzam.

**Visualization:** Mohammad T. Al Bataineh, Anas Alazzam.

**Writing – original draft:** Mohammad T. Al Bataineh, Anas Alazzam.

**Writing – review & editing:** Mohammad T. Al Bataineh, Anas Alazzam.

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
