## [Decision Letter · Decision Letter 0]

21 Jun 2023

PONE-D-23-16596Revolutionizing Medical Device Infection Control with Surface Treatment Using Microfabrication TechniquesPLOS ONE

Dear Dr. AL Bataineh,

Thank you for submitting your manuscript to PLOS ONE. After careful consideration, we feel that it has merit but does not fully meet PLOS ONE’s publication criteria as it currently stands. Therefore, we invite you to submit a revised version of the manuscript that addresses the points raised during the review process.

We look forward to receiving your revised manuscript.

Kind regards,

Geelsu Hwang, Ph.D.

Academic Editor

PLOS ONE

Journal Requirements:

Additional Editor Comments:

As shown in the reviewers' comments, there are lots of issues in the current manuscript to be addressed by the authors. Please address all the issues appropriately in the revised manuscript to be reconsidered for publication.

Reviewers' comments:

Reviewer's Responses to Questions

**Comments to the Author**

1. Is the manuscript technically sound, and do the data support the conclusions?

Reviewer #1: Partly

Reviewer #2: No

Reviewer #3: Partly

2. Has the statistical analysis been performed appropriately and rigorously? 

Reviewer #1: N/A

Reviewer #2: No

Reviewer #3: No

3. Have the authors made all data underlying the findings in their manuscript fully available?

Reviewer #1: Yes

Reviewer #2: Yes

Reviewer #3: No

4. Is the manuscript presented in an intelligible fashion and written in standard English?

Reviewer #1: No

Reviewer #2: Yes

Reviewer #3: Yes

5. Review Comments to the Author

Reviewer #1: Dear Authors,

This is an interesting study. However, the presentation is not acceptable. Poor English and lack of scientific presentation are lowering the merits of this study. It’s not acceptable in its current form. A major revision/rewriting is required for further consideration. I have some comments below for your reference.

Simple English should be used for better understanding. Readers who are not in your field should understand your manuscripts. Please maintain consistency in your writing. The aim and objective should be well reflected in the conclusion.

Redundant words are complicating this manuscript. Please avoid the unnecessary use of adjectives. Please check your English with a native proofreader.

Please use either medical “devices” or “apparatuses”. Don’t mix up.

The first appearance of the microbe’s name and its abbreviations should be consistent throughout the manuscript. The same should be applied to other abbreviations. Please check the entire manuscript.

Please follow the scientific way of writing. Materials and methods, results, and discussion should be presented scientifically. Please check the scientific nomenclature of microbes.

Methods: For indwelling medical device coating, a cytotoxicity test must be performed to justify its biocompatibility, and it is also good to assess hemocompatibility.

The result is the absolute findings of your experiments. No discussion and no reference should be added. In the discussion section, please discuss your result with references in a consistent manner. Please avoid a discussion that is out of context (not related to your experiment results).

Thanks

Reviewer #2: The authors study the adhesion and biofilm formation of C. albicans on a polymer, which is surface modified to acquire differences in surface wettability. The use of COC as a material to prevent C. albicans biofilm formation seem to be novel and the experiment design and the data to link surface wettability and Candida biofilm growth is interesting. However, it is not clear why several indwelling medical devices are tested, unless the authors had compared them with COC, which was not the case. It is also not clear what materials the medical devices are made of. Altogether, the authors can remove the information on the medical devices and focus a more rigorous study on the link between surface wettability and C. albicans biofilm formation using COC. The writing needs to be significantly improved, avoiding repetition of statements, a clear distinction between introduction, methods, results and discussion, and a logical flow of information making it easier to follow. The methods section should be more detailed and rigorous. Additionally, the study needs a clear hypothesis and also lack direction.

Please see below additional information

The authors seem to overplay the abstract. Keep it clear and simple and avoid words such as ‘remarkable’, ‘alarming’, ‘breakthrough finding’, and present the facts.

Please acknowledge in the introduction other instances where polymers are tested for wettability and Candida adhesion (ex: doi: 10.1128/AEM.71.12.8795-8801.2005)

The introduction can focus more on literature information about the impact of surface wettability on C. albicans biofilm development. Please find and summarize more related literature or state the lack of information and therefore, the need for studying it.

Line 71: Please write the full name of the bacteria on their first mention

Line 76: Candida need to be italicized throughout the manuscript and attention to detail.

Line 107: Need more information about fungal cells and their link with surface wettability/contact angles.

Line 120-124: Clearly state the surfaces the biofilms were developed. The methods need to be rewritten more clearly.

Line 135: What is the substrate used?

Figure 1, The schematic diagram and the caption can benefit from explaining upfront the aim of creating two different surfaces on the same substrate and clearly labelling which surface is hydrophobic and which is hydrophilic.

Line 154: O2 – 2 needs to be subscript

195-201: This paragraph can be removed from the results section and similarly avoid repetitive statements. Please present only the results here.

Line 194: This section does not add anything new to the manuscript. Please consider deleting this section. What is the substrate used in this experiment? Candida hyphae formation in the presence of serum and at 37C is well known.

Line 214-223: All this information belongs in the introduction.

Line 224-226: Need to describe the methods more clearly.

Line 228-231: After washing with DI water, only the hydrophilic area will hold a water droplet? Not clear how that works. Although photographs are shown (need to get more clear close-up photographs), this need to be further characterized and show there is no water in the hydrophilic area. How much can you tilt the substrate to prevent the water droplet from flowing/falling off the substrate? Need to show better ways of comparatively characterizing the wettability on the two surfaces.

Line 232: How was Candida grown on the substrate? Please explain this clearly in the methods section. How did you sterilize the substrate prior to Candida growth?

Line 256: Line 261-264: How did you perform the XTT assay on the same substrate (hydrophobic and hydrophilic areas)? This is not clearly explained in the methods section. Did you use two separate substrates for the XTT assay?

Figure 5C is not clear whether it has Candida cells. A light microscopy image will show the cells and hyphae clearly on each surface.

If you are proposing that the COC is a better polymer than the materials used for the commercial devices, then the level of Candida biofilm formation on all the surfaces from each device should be compared with COC. This needs to be quantified using XTT assays and qualitatively observed using microscopy. Additionally, each surface compared to COC should have been tested for surface wettability and Candida biofilm formation and explain the link between surface wettability and Candida biofilm formation. Currently, there is no apparent link between the COC and the medical implants that is tested in the manuscript.

Reviewer #3: The paper “Revolutionizing Medical Device Infection Control with Surface Treatment Using Microfabrication” describes using Cyclic olefin copolymer to coat plastic surfaces to evaluate Candida albicans biofilm growth. The authors microfabricated a wafer with patterned wettability, which can influence contact angle. The paper's central idea is how hydrophobicity could be used to avoid or reduce fungi biofilm. While the concept of controlling the wettability of surfaces to prevent biofilm growth has been demonstrated before, the anti-biofilm properties of available commercial medical devices are still an interesting topic. Based on the simplistic model utilized and essential biological questions raised and not answered through the manuscript, I have several concerns that are listed below:

1- The title might be modified once an infection model was not used/developed. Fungi biofilm control might be more appropriate for the title.

2- The introduction might be more focused on the polymer and the microfabrication techniques to prevent biofilm formation.

3- Statistics analysis should be used for this study. This is a critical part of a study that involves biofilm formation and growth. What are the tests utilized? Are the tests performed in triplicates? What are the p values of the data found?

4- The controls for this study can be better designed. If the comparison was the current medical devices, why did they not test them for biofilm growth?

5- In Figure 4, the authors affirmed that increased filamentation was encased within the ECM when cells were treated with serum at 37 °, compared to ECM-coated yeast cells at 30 o. However, the SEM images do not demonstrate significant differences in the biofilm structure.

6- Figure 5 describes the effect of surface wettability on promoting candida growth. However, any quantitative data was provided to affirm that.

7- In Figure 6, what are the significant differences between the groups?

6. PLOS authors have the option to publish the peer review history of their article (what does this mean?). If published, this will include your full peer review and any attached files.

Reviewer #1: No

Reviewer #2: No

Reviewer #3: No

---

## [Author Response · Author response to Decision Letter 0]

4 Aug 2023

Dear Professor Geelsu Hwang,

Thank you very much for handling the review of our manuscript titled “Revolutionizing Medical Device Infection Control with Surface Treatment Using Microfabrication Techniques” which we would like to be considered for publication in the prestigious journal, PLOS One. Please note that we took all reviewers' comments into considerations and have revised the manuscript accordingly. Below are our responses to all the issues raised by the reviewers as well as a summary of changes that have been incorporated into a revised version of our manuscript.

Reviewer #1: 

Dear Authors,This is an interesting study. However, the presentation is not acceptable. Poor English and lack of scientific presentation are lowering the merits of this study. It’s not acceptable in its current form. A major revision/rewriting is required for further consideration. I have some comments below for your reference.

Simple English should be used for better understanding. Readers who are not in your field should understand your manuscripts. Please maintain consistency in your writing. 

The aim and objective should be well reflected in the conclusion. Redundant words are complicating this manuscript. Please avoid the unnecessary use of adjectives. Please check your English with a native proofreader.

Response: We thank the reviewer for this feedback and for the time spent reading and evaluating our manuscript. In the revised manuscript, we simplified the language used when appropriate, reduced redundancy, modified conclusion to reflect the aims the study, and used native English proofreader.

Please use either medical “devices” or “apparatuses”. Don’t mix up.

Response: we updated the manuscript to use the term “medical devices” throughout the manuscript. 

The first appearance of the microbe’s name and its abbreviations should be consistent throughout the manuscript. The same should be applied to other abbreviations. 

Please check the entire manuscript.

Response: We thank the reviewer for this comment. We ensured consistent usage of the microbe's name and abbreviations throughout the revised manuscript, as well as maintaining uniformity with other abbreviations. 

Please follow the scientific way of writing. Materials and methods, results, and discussion should be presented scientifically. Please check the scientific nomenclature of microbes.

Response: We thank the reviewer for the suggestion. We updated the manuscript to restructure the manuscript as per the recommendation. 

Methods: For indwelling medical device coating, a cytotoxicity test must be performed to justify its biocompatibility, and it is also good to assess hemocompatibility.

Response: We sincerely appreciate the reviewer's insightful comment. The current study investigates the impact of wettability via plasma surface modification, eliminating the need for any additional coatings. Plasma treatment of medical devices is a widely recognized technique within the medical field, and as such, the authors believe that conducting a cytotoxicity test to confirm biocompatibility is not applicable to this study. 

The result is the absolute findings of your experiments. No discussion and no reference should be added. 

Response: We thank the reviewer for the comment. The manuscript has been updated following the reviewer’s comment. 

In the discussion section, please discuss your result with references in a consistent manner. 

Please avoid a discussion that is out of context (not related to your experiment results).

Thanks

Response: L322-328 removed. 

Reviewer #2: The authors study the adhesion and biofilm formation of C. albicans on a polymer, which is surface modified to acquire differences in surface wettability. The use of COC as a material to prevent C. albicans biofilm formation seem to be novel and the experiment design and the data to link surface wettability and Candida biofilm growth is interesting. However, it is not clear why several indwelling medical devices are tested, unless the authors had compared them with COC, which was not the case. It is also not clear what materials the medical devices are made of. 

Altogether, the authors can remove the information on the medical devices and focus a more rigorous study on the link between surface wettability and C. albicans biofilm formation using COC. 

We express our gratitude to the reviewer for their valuable comment, and we extend our apologies for any lack of clarity regarding the rationale for incorporating wettability measurements into the study of commercial indwelling devices. Our objective in conducting surface contact angle tests on these medical devices was to gain insights into the wettability of their internal surfaces. 

The outcomes of our investigation revealed that all tested devices exhibited hydrophilic surfaces, which can be surely assumed to create a conducive environment for the attachment of C. albicans, consistent with numerous clinical reports of increased resistance due to indwelling devices and as demonstrated in the findings presented within this manuscript with respect to COC. We have taken utmost care to elucidate this point thoroughly in the discussion section of the revised manuscript.

The writing needs to be significantly improved, avoiding repetition of statements, a clear distinction between introduction, methods, results and discussion, and a logical flow of information making it easier to follow.

Response: Thank you for your feedback. We enhanced the writing of the revised manuscript by minimizing repetition, ensuring clear section distinctions, and improving the logical flow for better readability.

The methods section should be more detailed and rigorous. Additionally, the study needs a clear hypothesis and also lack direction.

Response: We appreciate your valuable feedback. We enhanced the methods section in the revised manuscript with more information. Furthermore, we established a clear hypothesis and improved the overall direction of the research for greater clarity and focus. 

Please see below additional information

The authors seem to overplay the abstract. Keep it clear and simple and avoid words such as ‘remarkable’, ‘alarming’, ‘breakthrough finding’, and present the facts.

Response: In response to the reviewer's feedback, we have revised the abstract in the updated manuscript to exclude terms such as 'remarkable', 'alarming', and 'breakthrough finding'. Instead, we present only factual information.

Please acknowledge in the introduction other instances where polymers are tested for wettability and Candida adhesion (ex: doi: 10.1128/AEM.71.12.8795-8801.2005)The introduction can focus more on literature information about the impact of surface wettability on C. albicans biofilm development. 

Response: We appreciate your valuable suggestion. In the introduction, we acknowledged other instances where polymers were tested for wettability and Candida adhesion, including the reference you provided (doi: 10.1128/AEM.71.12.8795-8801.2005). Furthermore, we focused more on literature information regarding the impact of surface wettability on C. albicans biofilm development, thereby enhancing the contextual background of our study. 

Please find and summarize more related literature or state the lack of information and therefore, the need for studying it.

Response: The revised manuscript now includes the rationale for conducting this study, stemming from the limited existing literature on the subject. Moreover, it emphasizes the imperative to explore the influence of wettability on C. albicans biofilm formation.

Line 71: Please write the full name of the bacteria on their first mention

Response: Corrected in the revised manuscript. 

Line 76: Candida need to be italicized throughout the manuscript and attention to detail.

Response: Thank you for pointing this out. This was corrected throughout the revised manuscript. 

Line 107: Need more information about fungal cells and their link with surface wettability/contact angles.

Response: Attachment in Candida is governed by several adhesins, including Als1, Als3, Hwp1, and Eap1. These adhesins have been shown to play critical roles in adhesion and biofilm formation. In addition to adhesins, C. albicans also produce ECM components, such as β-glucan, chitin, and mannan, which are important for biofilm formation and stability and help to protect the cells within it from host defenses and antifungal agents. In our case, hydrophobic surfaces have been observed to induce a conformational change in fibronectin, resulting in a decrease in cell adhesion. Fibronectin is one of the major mechanisms for adhesion to the fungal cell wall. This is now modified in the introduction and parts of the discussion. 

Line 120-124: Clearly state the surfaces the biofilms were developed. The methods need to be rewritten more clearly.

Response: Biofilm growth is exclusively visible within the hydrophilic region as shown in Figure 6 and we added a new section under methods for biofilm formation. 

Line 135: What is the substrate used? 

Response: Cyclic olefin copolymer (COC).

Figure 1, The schematic diagram and the caption can benefit from explaining upfront the aim of creating two different surfaces on the same substrate and clearly labelling which surface is hydrophobic and which is hydrophilic.

Response: The concept behind incorporating two distinct regions of varying wettability on a single surface is to explore the hypothesis regarding the impact of wettability on C. albicans biofilm formation. This notion is visually represented within the manuscript through the inclusion of Figure 5, demonstrating the presence of these distinct regions on the same surface and modifying Figure 5 legend accordingly. 

Line 154: O2 – 2 needs to be subscript

Response: This was corrected in the revised manuscript. 

195-201: This paragraph can be removed from the results section and similarly avoid repetitive statements. Please present only the results here.

Response: The paragraph was removed from the revised manuscript following the reviewer’s recommendation. 

Line 194: This section does not add anything new to the manuscript. Please consider deleting this section. What is the substrate used in this experiment? Candida hyphae formation in the presence of serum and at 37C is well known.

Response: While we concur with the reviewer that the added information did not introduce novel insights, we included this section to validate the successful biofilm formation in our YEPD.

214-223: All this information belongs in the introduction.

Response: The mentioned lines were moved to the introduction part of the revised manuscript. 

Line 224-226: Need to describe the methods more clearly.

Response We modified two distinct wettabilities using microfabrication and plasma treatment to selected surface areas and for more details ref doi: 10.3390/polym13142305. 

Line 228-231: After washing with DI water, only the hydrophilic area will hold a water droplet? Not clear how that works. Although photographs are shown (need to get more clear close-up photographs), this need to be further characterized and show there is no water in the hydrophilic area. 

How much can you tilt the substrate to prevent the water droplet from flowing/falling off the substrate? Need to show better ways of comparatively characterizing the wettability on the two surfaces.

Response: Wettability significantly influences the attachment behavior of water droplets to surfaces. Hydrophilic surfaces encourage water to spread with a small contact angle (<90 degrees), while hydrophobic surfaces cause droplets to bead up with a larger angle (>90 degrees). This attachment process relies on cohesive and adhesive forces, bearing relevance for applications like self-cleaning coatings and biofilm formation. Notably, the authors previously published a study on modifying COC surface wettability using plasma and graphene oxide coating (doi: 10.3390/polym13142305), offering further insights into the COC substrate's wettability.

Surface energy, a key factor, governs the strength of attraction between the surface and water droplets, impacting wetting and spreading. Higher surface energy promotes greater wetting, while lower surface energy results in beading and reduced spreading. The accompanying figure illustrates a COC surface with varied wettabilities, transitioning from high surface energy (where droplets spread) to low surface energy (where droplets do not spread).

In hydrophobic scenarios, water easily falls off the surface. For hydrophilic cases, water falling depends on the interplay of forces between the droplet and the surface.

Line 232: How was Candida grown on the substrate? Please explain this clearly in the methods section. 

Response: The candida culture was diluted at a ratio of 1:10 into 50 ml of pre-warmed YEPD medium supplemented with 10% fetal bovine serum (FBS) at 37°C. The resulting cultures were agitated at 200 rpm for a duration of 24 hours, following previously established protocols. 

How did you sterilize the substrate prior to Candida growth?

Response: The substrates underwent sterilization by being washed with IPA (Isopropyl Alcohol) and modified in line 

Line 256: Line 261-264: How did you perform the XTT assay on the same substrate (hydrophobic and hydrophilic areas)? This is not clearly explained in the methods section. Did you use two separate substrates for the XTT assay?

Response: We modified the methods section to explain that we used two separate substrates for the XTT assay. 

Figure 5C is not clear whether it has Candida cells. A light microscopy image will show the cells and hyphae clearly on each surface.

Response: We agree, kindly see the magnified light microscopy, below and can also be seen in Figure 6. 

If you are proposing that the COC is a better polymer than the materials used for the commercial devices, then the level of Candida biofilm formation on all the surfaces from each device should be compared with COC. This needs to be quantified using XTT assays and qualitatively observed using microscopy. Additionally, each surface compared to COC should have been tested for surface wettability and Candida biofilm formation and explain the link between surface wettability and Candida biofilm formation. Currently, there is no apparent link between the COC and the medical implants that is tested in the manuscript.

Response: We agree. In this study, we reported COC wettability modifications as a promising approach to modify biofilm formation. In this study, we presented an assessment of the wettability of commercially available indwelling devices. However, it's important to note that our focus did not encompass the investigation of biofilm formation on these devices. The selection of COC as our substrate stemmed from its suitability, owing to the authors' established approach for patterning its wettability and its transparent nature, which facilitates the visualization of yeast on its surface. We acknowledge the points raised by the reviewer as limitations to be addressed in the follow-up study. 

Reviewer #3: The paper “Revolutionizing Medical Device Infection Control with Surface Treatment Using Microfabrication” describes using Cyclic olefin copolymer to coat plastic surfaces to evaluate Candida albicans biofilm growth. The authors microfabricated a wafer with patterned wettability, which can influence contact angle. The paper's central idea is how hydrophobicity could be used to avoid or reduce fungi biofilm. While the concept of controlling the wettability of surfaces to prevent biofilm growth has been demonstrated before, the anti-biofilm properties of available commercial medical devices are still an interesting topic. Based on the simplistic model utilized and essential biological questions raised and not answered through the manuscript, I have several concerns that are listed below:

1-The title might be modified once an infection model was not used/developed. Fungi biofilm control might be more appropriate for the title.

Response: We appreciate your valuable suggestion and thank you for your time and effort, we modified the title accordingly. 

2- The introduction might be more focused on the polymer and the microfabrication techniques to prevent biofilm formation.

Response: modified, lines 118 to 121 

3- Statistics analysis should be used for this study. This is a critical part of a study that involves biofilm formation and growth. What are the tests utilized? Are the tests performed in triplicates? What are the p values of the data found?

Response: We added the following under Statistical analyses. Each experiment was run three times with triplicates each time. A P value of less than 0.05 was regarded as statistically significant for all statistical tests (unpaired t test and correlation coefficient).

4- The controls for this study can be better designed. If the comparison was the current medical devices, why did they not test them for biofilm growth?

Response: We express our gratitude to the reviewer for their valuable comment, and we extend our apologies for any lack of clarity regarding the rationale for incorporating wettability measurements into the study of commercial indwelling devices. Our objective in conducting surface contact angle tests on these medical devices was to gain insights into the wettability of their internal surfaces. 

The outcomes of our investigation revealed that all tested devices exhibited hydrophilic surfaces, which can be surely assumed to create a conducive environment for the attachment of C. albicans, consistent with numerous clinical reports of increased resistance due to indwelling devices and as demonstrated in the findings presented within this manuscript with respect to COC. We have taken utmost care to elucidate this point thoroughly in the discussion section of the revised manuscript.

5- In Figure 4, the authors affirmed that increased filamentation was encased within the ECM when cells were treated with serum at 37 °, compared to ECM-coated yeast cells at 30 o. However, the SEM images do not demonstrate significant differences in the biofilm structure.

Response: We agree with the reviewer, however the dimorphic switching under serum and at 37◦C is well documented in the literature. Our goal here is to confirm the successful biofilm formation in YEPD. Furthermore, we changed Figure 4 to reflect a more significant difference in the biofilm structure available to the visual field. This was also confirmed by XTT assay. 

6- Figure 5 describes the effect of surface wettability on promoting candida growth. However, any quantitative data was provided to affirm that.

Response: We agree and XTT assay is severe as a semi-quantitative method. 

7- In Figure 6, what are the significant differences between the groups?

Response: P <0.05

---

## [Decision Letter · Decision Letter 1]

30 Aug 2023

PONE-D-23-16596R1Revolutionizing Medical Device Biofilm Control with Surface Treatment Using Microfabrication TechniquesPLOS ONE

Dear Dr. AL Bataineh,

Thank you for submitting your manuscript to PLOS ONE. After careful consideration, we feel that it has merit but does not fully meet PLOS ONE’s publication criteria as it currently stands. Therefore, we invite you to submit a revised version of the manuscript that addresses the points raised during the review process.

We look forward to receiving your revised manuscript.

Kind regards,

Geelsu Hwang, Ph.D.

Academic Editor

PLOS ONE

Additional Editor Comments:

Dear authors,

One of the reviewers still expressed concerns about your article. Please review the comments, address them carefully, and submit the revised manuscript as soon as possible.

Reviewers' comments:

Reviewer's Responses to Questions

**Comments to the Author**

1. If the authors have adequately addressed your comments raised in a previous round of review and you feel that this manuscript is now acceptable for publication, you may indicate that here to bypass the “Comments to the Author” section, enter your conflict of interest statement in the “Confidential to Editor” section, and submit your "Accept" recommendation.

Reviewer #1: All comments have been addressed

Reviewer #2: (No Response)

2. Is the manuscript technically sound, and do the data support the conclusions?

Reviewer #1: Partly

Reviewer #2: Partly

3. Has the statistical analysis been performed appropriately and rigorously? 

Reviewer #1: (No Response)

Reviewer #2: Yes

4. Have the authors made all data underlying the findings in their manuscript fully available?

Reviewer #1: Yes

Reviewer #2: Yes

5. Is the manuscript presented in an intelligible fashion and written in standard English?

Reviewer #1: Yes

Reviewer #2: No

6. Review Comments to the Author

Reviewer #1: It is better now and can be accepted for publication. No further comments. I suggest having a pre-publication proofreading.

Reviewer #2: Reviewers' Comments are based on the numbering of the version with the tracked changes.

Line 1: Topic should be changed to read better. Please consider removing ‘Revolutionizing”.

Line 37: found to be?

Line 39: Where was the Candida biofilm growth established? If it is on the medical devices, please mention it.

I don’t see any mention about the COC. The data on COC takes up a significant portion of the manuscript and therefore, must be presented in the abstract. The abstract can be better written.

Line 42: C. albicans repetition

Line 73: Still the full scientific name of the microorganisms are not mentioned in the first instance. Here and throughout the manuscript, please pay attention to detail.

In the introduction, need to introduce COC and the rationale for using it to test the wettability.

This reviewer thinks the introduction can be trimmed and need more work to give it more focus.

Methods:

Line 139: The sentence should start by saying “For imaging purposes”? Otherwise, why is it treated with formaldehyde? The methods need to be very clear and logically flow throughout the manuscript.

Line 143-152: The writing is very confusing. This reviewer believes the authors can do a better job in clearly explaining the methodology, here and throughout the methods section. 2,3-bis(2-methoxy-4-nitro-5-sulfophenyl) is mentioned at the beginning and then XTT reagent at the end of the paragraph. Perhaps stick to one chemical nomenclature? Spelling mistakes: XXT, 24 hr. or 24 h. or 24 h please be consistent throughout the manuscript. Need attention to detail.

Statistical analysis section can be the final sub topic in the methods section.

Figure 1E: Can the authors show the hydrophilic and hydrophobic regions in the disc (in the acetone bath) in this image as well, similar to the other ones.

Line 188-191: Do the authors think these sentences belong in the results section together with Figure 2? Perhaps a comparison with the water contact angle measurements of the medical devices? This would be more appropriate.

Line 210-211: What are materials that the medical devices are made of? This need to be mentioned.

Line 223: Either remove this section or It is better to evaluate the C. albicans biofilm formation on each medical devices (and perhaps if you can compare them with COC - hydrophobic and hydrophilic surfaces)? Perhaps do a XTT assay? This will be a good dataset to compare with the above water contact angle measurements. This will also be a good rationale to further study Wettability and C. albicans biofilm formation on COC in the sections below. And Figure 4 can be moved to a supplementary file.

Line 298 : XXT?

Line 244 and Line 275: Change the topic of these two sections. Currently both are very similar.

Figure 5 and 6: Can you show better quality images, especially Figure 5B and 5C? Also some labelling/indicators (hydrophobic and hydrophilic regions, water droplets, C. albicans growth) would help the readers. Although the information is indicated in the figure caption, labeling some of these information will bring more clarity.

Line 342-343: The study did not prove the involvement of proteins in the attachment. This statement must be removed.

Line 346-349: These sentences can be trimmed.

Line 362: XXT ? Spelling mistakes and typos here and throughout the manuscript. Need more attention to detail.

The story can still be made to flow better with logical flow of information and clear writing. This reviewer believes that the manuscript can still be improved.

7. PLOS authors have the option to publish the peer review history of their article (what does this mean?). If published, this will include your full peer review and any attached files.

Reviewer #1: No

Reviewer #2: No

---

## [Author Response · Author response to Decision Letter 1]

19 Sep 2023

Response to Reviewers

Reviewer #1: It is better now and can be accepted for publication. No further comments. 

I suggest having a pre-publication proofreading.

Response: We thank Reviewer #1 for time and constructive feedback, and performed the suggested proofreading. 

Reviewer #2: Reviewers' Comments are based on the numbering of the version with the tracked changes.

Line 1: Topic should be changed to read better. Please consider removing ‘Revolutionizing”.

Response: modified as requested. 

Line 37: found to be?

Response: Yes, and modified as requested. 

Line 39: Where was the Candida biofilm growth established? If it is on the medical devices, please mention it. I don’t see any mention about the COC. The data on COC takes up a significant portion of the manuscript and therefore, must be presented in the abstract. The abstract can be better written.

Response: We agree, and modified as requested.

Line 42: C. albicans repetition

Response: Corrected. 

Line 73: Still the full scientific name of the microorganisms are not mentioned in the first instance. Here and throughout the manuscript, please pay attention to detail.

Response: Corrected.

In the introduction, need to introduce COC and the rationale for using it to test the wettability. This reviewer thinks the introduction can be trimmed and need more work to give it more focus.

Response: The introduction part of the revised manuscript has been updated as suggested.

Methods: Line 139: The sentence should start by saying “For imaging purposes”? Otherwise, why is it treated with formaldehyde? The methods need to be very clear and logically flow throughout the manuscript.

Response: Modified as suggested.

Line 143-152: The writing is very confusing. This reviewer believes the authors can do a better job in clearly explaining the methodology, here and throughout the methods section. 2,3-bis(2-methoxy-4-nitro-5-sulfophenyl) is mentioned at the beginning and then XTT reagent at the end of the paragraph. Perhaps stick to one chemical nomenclature? Spelling mistakes: XXT, 24 hr. or 24 h. or 24 h please be consistent throughout the manuscript. Need attention to detail.

Response: Modified as suggested.

Statistical analysis section can be the final sub topic in the methods section.

Response: Modified as suggested.

Figure 1E: Can the authors show the hydrophilic and hydrophobic regions in the disc (in the acetone bath) in this image as well, similar to the other ones.

Response: Figure 1 shows the fabrication technique used to pattern the wettability of the COC surface. The hydrophobic and hydrophilic properties are not shown here. Figure 1E shows the wafer in acetone to perform liftoff, the hydrophilic/hydrophobic properties can’t be shown in the figure. 

Line 188-191: Do the authors think these sentences belong in the results section together with Figure 2? Perhaps a comparison with the water contact angle measurements of the medical devices? This would be more appropriate.

Response: The wettability of the COC substrate was previously addressed in our earlier publication and is not the central focus of the current manuscript. The authors firmly believe that the placement of these lines within the method section of the manuscript is appropriate, and they should not be considered as part of the results section.

Line 210-211: What are materials that the medical devices are made of? This need to be mentioned.

Response: The materials used and the sterilization method for the four medical devices are

• Butterfly Cannula tubing is made from PVC and sterilized using irradiation.

• Thoracic Catheter is made from clear thermosensitive PVC and sterilized using Ethylene oxide gas. 

• Ryle’s Tube is made from medical grade PVC and sterilized by Ethylene oxide gas. 

• Foley Catheter is made from latex and sterilized by Ethylene oxide gas.

The above information has been added to Figure 3 of the revised manuscript.

Line 223: Either remove this section or It is better to evaluate the C. albicans biofilm formation on each medical devices (and perhaps if you can compare them with COC - hydrophobic and hydrophilic surfaces)? Perhaps do a XTT assay? This will be a good dataset to compare with the above water contact angle measurements. This will also be a good rationale to further study Wettability and C. albicans biofilm formation on COC in the sections below. And Figure 4 can be moved to a supplementary file.

Response: We agree, and have acknowledged this as a limitation of the current study and have indicated this in the discussion L354. 

Line 298 : XXT? 

Response: yes and corrected.

Line 244 and Line 275: Change the topic of these two sections. Currently both are very similar.

Response: we agree, and changed the topics accordingly.

Figure 5 and 6: Can you show better quality images, especially Figure 5B and 5C? Also some labelling/indicators (hydrophobic and hydrophilic regions, water droplets, C. albicans growth) would help the readers. Although the information is indicated in the figure caption, labeling some of these information will bring more clarity.

Response: Figure 5 is updated to include labelling and information about the three subfigures. The caption is also updated to include information about the figure. 

Line 342-343: The study did not prove the involvement of proteins in the attachment. This statement must be removed.

Response: We agree, and removed.

Line 346-349: These sentences can be trimmed.

Response: Modified as suggested.

Line 362: XXT ? Spelling mistakes and typos here and throughout the manuscript. Need more attention to detail.

Response: corrected.

The story can still be made to flow better with logical flow of information and clear writing. This reviewer believes that the manuscript can still be improved.

Response: The revised manuscript has diligently incorporated all of the suggested changes made by the reviewer. These modifications have greatly enhanced the overall flow of ideas and provided a significant boost in clarity. We extend our heartfelt gratitude to the reviewer for their invaluable and constructive feedback.

---

## [Decision Letter · Decision Letter 2]

27 Sep 2023

Transforming Medical Device Biofilm Control with Surface Treatment Using Microfabrication Techniques

PONE-D-23-16596R2

Dear Dr. AL Bataineh,

We’re pleased to inform you that your manuscript has been judged scientifically suitable for publication and will be formally accepted for publication once it meets all outstanding technical requirements.

Kind regards,

Geelsu Hwang, Ph.D.

Academic Editor

PLOS ONE

Reviewers' comments:

Reviewer's Responses to Questions

**Comments to the Author**

1. If the authors have adequately addressed your comments raised in a previous round of review and you feel that this manuscript is now acceptable for publication, you may indicate that here to bypass the “Comments to the Author” section, enter your conflict of interest statement in the “Confidential to Editor” section, and submit your "Accept" recommendation.

Reviewer #2: All comments have been addressed

2. Is the manuscript technically sound, and do the data support the conclusions?

Reviewer #2: Yes

3. Has the statistical analysis been performed appropriately and rigorously? 

Reviewer #2: Yes

4. Have the authors made all data underlying the findings in their manuscript fully available?

Reviewer #2: Yes

5. Is the manuscript presented in an intelligible fashion and written in standard English?

Reviewer #2: Yes

6. Review Comments to the Author

Reviewer #2: Overall much improved. A spell check need to be performed and check for typos. A final proof read would help.

7. PLOS authors have the option to publish the peer review history of their article (what does this mean?). If published, this will include your full peer review and any attached files.

Reviewer #2: No

---

## [Editor Report · Acceptance letter]

4 Oct 2023

PONE-D-23-16596R2 

Transforming Medical Device Biofilm Control with Surface Treatment Using Microfabrication Techniques 

Dear Dr. Al Bataineh:

I'm pleased to inform you that your manuscript has been deemed suitable for publication in PLOS ONE. Congratulations! Your manuscript is now with our production department. 

Kind regards, 

on behalf of

Dr. Geelsu Hwang 

Academic Editor

PLOS ONE